# Neuropsychiatric Consequences of Lipophilic Beta-Blockers

**DOI:** 10.3390/medicina57020155

**Published:** 2021-02-09

**Authors:** Sabina Alexandra Cojocariu, Alexandra Maștaleru, Radu Andy Sascău, Cristian Stătescu, Florin Mitu, Maria Magdalena Leon-Constantin

**Affiliations:** 1Department of Medical Specialties (I), Faculty of Medicine, “Grigore T Popa” University of Medicine and Pharmacy, University Street nr 16, 700115 Iasi, Romania; sabina_cojocariu@yahoo.com (S.A.C.); radu.sascau@gmail.com (R.A.S.); cstatescu@gmail.com (C.S.); mitu.florin@yahoo.com (F.M.); leon_mariamagdalena@yahoo.com (M.M.L.-C.); 2Clinical Rehabilitation Hospital–Cardiovascular Rehabilitation Clinic, Pantelimon Halipa Street nr 14, 700661 Iasi, Romania; 3Institute of Cardiovascular Disease “Prof. Dr. George. I.M. Georgescu”, Carol I Boulevard nr 50, 700503 Iasi, Romania

**Keywords:** beta-blockers, depression, fatigue, propranolol, migraine

## Abstract

Beta-blockers are a class of drugs with important benefits in cardiovascular pathology. In this paper, we aim to highlight their adverse and therapeutic effects in the neuropsychiatric field. With respect to permeability, we would like to mention that most beta-blockers are lipophilic and can cross the blood–brain barrier. Observational studies show the presence of neuropsychiatric side effects when taking beta-blockers, and is the reason for which caution is recommended in their use in patients with depressive syndrome. From a therapeutic point of view, most current evidence is for the use of beta-blockers in migraine attacks, essential tremor, and akathisia. Beta-blockers appear to be effective in the treatment of aggressive behavior, beneficial in the prevention of posttraumatic stress syndrome and may play a role in the adjuvant treatment of obsessive–compulsive disorder, which is refractory to standard therapy. In conclusion, the relationship between beta-blockers and the central nervous system appears as a two-sided coin. Summarizing the neuropsychiatric side effects of beta-blockers, we suggest that clinicians pay special attention to the pharmacological properties of different beta-blockers.

## 1. Introduction

Beta-blockers are widely used for the treatment of cardiovascular and non-cardiovascular conditions [1,2,3]. Studies have shown that about 30% of the elderly are on beta-blocker treatment, hence the need to deepen the pharmacokinetic features [4]. The neuropsychiatric and cognitive effects of beta-blockers were proven many years ago. However, the evidence is limited by the use in trials of pharmaceutical preparations or different doses, but also by the application of different questionnaires of cognitive assessment [5,6,7]. The side effects on the central nervous system are heterogeneous, difficult to measure in clinical trials, and often require a prolonged follow-up. Both neuropsychiatric side effects and primary neurodegenerative diseases are common in patients with cardiovascular disease, making it difficult to attribute the symptoms to a concrete cause [8,9,10,11]. Therefore, this paper will summarize the pharmacokinetic and pharmacodynamic peculiarities of beta-blockers in relation to neuropsychiatric effects.

The lipophilic property of some beta-blockers is the ability of the drug to diffuse through the blood–brain barrier [12]. First, the agent must cross it and attach directly to the beta-adrenergic receptors, the suppression of which it mediates. Second, the drug must also interact with non-adrenergic receptors, blocking their signals or destabilizing the cell membrane [13]. Highly lipophilic beta-blockers, such as propranolol, diffuse rapidly through the brain tissue compared to hydrophilic beta-blockers, such as atenolol, which lacks this property [3].

## 2. Research Strategy

For this review, we employed the search strategy from well-known internationally accepted databases, namely PubMed, ScienceDirect, and Web of Science, using specific keywords, including beta-blockers side effects and psychiatric disorders associated with this medication. For investigation, a combination of keywords was used [beta-blockers; propranolol; metoprolol; atenolol; timolol; pindolol] + [fatigue; depression; sleep disorders; nightmares; hallucinations; delirium; Parkinson’s disease; risk of falling; migraine; tremor; akathisia; anxiety; posttraumatic stress syndrome; aggression; obsessive–compulsive disorder] + liposolubility. We included relevant reviews, meta-analyses, case reports, case studies, and comparative studies in the search strategy. When we started the research strategy, we discovered the lack of articles published in the last couple of years, this fact forcing us to include both articles dating back to 1975 up until 2020. Our research included prospective and retrospective cohorts and randomized clinical trials that included patients in treatment with lipophilic beta-blockers that had at least one neuropsychiatric symptoms (fatigue, depression, sleep disorders and nightmares, hallucinations, delirium, or other central nervous system therapeutic effects, such as migraine, tremor, akathisia, anxiety, aggression, or obsessive–compulsive disorder). Studies with patients under 18 years old, patients in treatment with hydrophilic beta-blockers, studies that did not mention or have psychiatric side effects, or small studies that included less than 50 patients were excluded. Beta-blockers are extremely frequently used in patients with cardiovascular disease, given it is recommended by all the guidelines, but we considered it useful to draw attention to their side effects, especially the ones in the psychiatric sphere. For example, a search with the keywords “beta-blockers side effects depression” yields 5 titles published in 2020, 14 titles in 2019, 6 in 2018, 9 in 2017, and 5 in 2016.

## 3. Pharmacological Characteristics

The physicochemical properties of molecules are of significant importance for therapeutic and side effects [14]. Although beta-blockers have similar pharmacotherapeutic effects, they may have different pharmacokinetic properties: gastrointestinal absorption rate, first-pass effect, lipid solubility, hepatic biotransformation rate, the pharmacological activity of metabolites, and clearance rate [15]. According to these pharmacokinetic properties, beta-blockers are divided into two major groups: agents eliminated from the body by hepatic metabolism and those excreted unchanged by the kidneys. First-class beta-blockers, such as propranolol or metoprolol, are almost completely absorbed in the small intestine, metabolized by the liver, and have a high solubility in lipids. In contrast, beta-blockers in the other category, such as atenolol or sotalol, are less or not at all soluble in lipids, incompletely absorbed and excreted unchanged by the kidney [16]. Beta-blockers were classified according to three fundamental characteristics for the understanding of the effects on the central nervous system: liposolubility, intrinsic sympathomimetic activity, and cardioselectivity (Table 1).

Liposolubility is defined as the physicochemical property of a substance that determines the fundamental characteristics of a medicinal preparation, with an impact on the metabolic, pharmacokinetic, pharmacodynamic, and toxicological profiles. Currently, numerous clinical trials have reported a significant relationship between permeability and lipophilia. The more liposoluble and smaller the substance, the more permeable it is through biological membranes [17]. Lipid solubility is not the only factor responsible for central nervous system effects. Intrinsic sympathomimetic activity, also known as partial agonist activity, allows the agent a minimum degree of beta-stimulation at rest [18]. Cardioselectivity refers to the ability of a pharmacological agent to block beta-1 adrenergic receptors preferentially. This property is relative; selective beta-blockers used in high doses can also cause inhibition of beta-2 adrenergic receptors [19].

Liposolubility is the ability of a biochemical compound to dissolve in fats, oils, lipids, and nonpolar solvents, such as hexane or toluene. In vivo, it is the key element for the diffusion through the cell membrane and binding to protein sites. Most often, lipophilia overlaps with the term hydrophobicity used to describe nature, both of which are used to describe the force of dispersion. There are particular situations in which the two terms do not have the same meaning; compounds that are hydrophobic but not lipophilic: silicone, fluorocarbons [20]. Lipophilia can be measured by determining the partition coefficient (log P), a molecular parameter that describes the balance of a non-ionized solute between water and an immiscible organic solvent [21]. Data from the literature reported by Lipinski et al. show that a substance with log P < 5 has the characteristics of a lipophilic molecule [22]. At present, current evidence shows that a value of >1 and <3 of the partition coefficients provide a substance with appropriate physicochemical properties for therapeutic success and limited side effects [23]. The difference between lipophilic and hydrophilic beta-blockers can be observed in Figure 1.

The distribution of a drug in the central nervous system can also be explained by the structure of the interface between the blood system and the cerebrospinal system [24]. The blood–brain barrier is composed of endothelial vascular cells, astrocytes, and pericytes based in the choroid plexus, cerebral vascularity, and ependyma [24,25]. Capillary endothelial cells are interconnected by tight junctions containing efflux pumps (P-glycoprotein) with a role in limiting membrane permeability [26]. In general, small and highly lipophilic molecules can passively cross cell membranes that form the blood–brain barrier [27]. On the other hand, active transport mechanisms can influence the movement of the biological barrier to one side or the other; substances with high lipophilia have an increased affinity for certain efflux pumps [17]. Hence, getting across the blood–brain barrier is associated with various central nervous system consequences, as can be seen in Figure 2.

## 4. Central Nervous System Side Effects

Although beta-blockers have a high benefit/risk ratio, in some situations, they can negatively affect the quality of life through side effects. In general, side effects are reduced over time, are more common at high doses, and tend to be more frequent in elderly patients. By blocking beta-adrenergic receptors, the most common side effects are fatigue and cold extremities. Over time, beta-blocker treatment has been associated with a number of consequences for the central nervous system: fatigue, depression, sleep disorders and nightmares, visual hallucinations, delirium or psychosis, Parkinson’s disease, and the risk of falling [28].

### 4.1. Fatigue

Fatigue is a subjective clinical manifestation expressed by a deficiency of mental or physical energy. The mechanism of production is complex and poorly understood at present, involving both the central and the peripheral nervous systems [29,30,31]. Fatigue depends on intrinsic and extrinsic factors, such as the patient’s general mood or weather situation [32]. The Na^+^/K^+^-ATPase pump that controls the movement of ions between muscle cells and plasma may be one of the pathogenic pathways involved in the development of fatigue in patients treated with beta-blockers [33]. Both cardio selective and first-generation beta-blockers have a higher rate of association with fatigue compared to non-selective and new-generation beta-blockers [34]. The duration of treatment has a significant influence. Acute treatment with beta-blockers increases the risk of inducing fatigue and is often poorly tolerated by the patient, which can lead to therapeutic noncompliance. In contrast, in hypertensive patients undergoing beta-blocker treatment for long periods of time, the fatigue seems to disappear spontaneously [35].

### 4.2. Depression

Although the association between the use of beta-blocker treatment and the development of depression has been widely described over time, it is still a controversial topic nowadays. Several case reports and small general reports have suggested a relationship between propranolol and depression [28]. Specialists have proposed several mechanisms to explain a possible association between beta-blockers and depression in elderly patients, both through side effects regarding the sleep and circadian cycle regulation, as well as through sympathetic mediated feedback [36]. Two clinical trials have shown that antidepressants are prescribed at a significant rate in patients treated with propranolol [37,38].

On the contrary, a study on 312 patients receiving propranolol treatment did not show a substantial association with depressive symptoms after one year of follow-up [39]. A comprehensive overview of 5800 patients showed that the use of propranolol predisposes to depression in rare situations and that such symptoms occur after prolonged administration [40]. The most extensive analysis in relation to the association between beta-blockers and depression, a meta-analysis of 15 clinical trials (over 35,000 patients), showed that there was no significant increase in the incidence of depressive symptoms [34]. Jin et al. followed the 1-year incidence of major depressive disorder in patients without a history of antidepressant treatment or a diagnosis of depression in which beta-blocker was initiated for cardiovascular disease. Recently, the results of an observational study were published that showed the fact that the use of non-selective beta-blockers, female sex, a history of anxiety, and common chronic diseases (gastrointestinal and musculoskeletal) were predictive [41].

In patients with heart failure, data from the literature report an 11–45% prevalence of adverse events, such as anxiety and depression, in the cohort of patients undergoing beta-blocker treatment [42]. In the case of patients with chronic coronary syndrome, a study by Burkauskas et al. demonstrated an association between beta-blocker treatment and psychological symptoms [43]. Emotional stress caused by anxiety or depression can degenerate into a burnout syndrome clinically expressed through fatigue and exhaustion [1].

### 4.3. Sleep Disorders and Nightmares

The sympathetic vegetative nervous system is involved in inducing and maintaining sleep, while serotonin plays an important role in the physiology of normal sleep [44]. Beta-blockers, by their action of antagonizing beta-adrenergic and serotonergic receptors, suppress rapid eye movement (REM) sleep and can induce disorders up to insomnia [45]. Patients with insomnia have difficulty inducing sleep, maintaining sleep, or early morning awakening without the ability to fall back asleep [46]. Chronic insomnia occurs in 9–15% of patients with sleep disorders and can lead to significant daytime consequences, such as fatigue, sleepiness, inattention, concentration deficiency, impaired performance, and mood disturbance [47].

Melatonin plays a key role in regulating the circadian rhythm and sleep–wake cycle. The synthesis and secretion of melatonin in the systemic circulation are influenced by norepinephrine through beta 1-adrenergic receptors. Beta-blockers reduce the level of melatonin in the body and can thus induce sleep disorders and nightmares [48]. Nightmares are dysphoric dreams that generally occur during REM sleep in the last third of the night. Nightmares become pathological when they are frequent and disabling, affecting the individual from a social, occupational, and emotional point of view [44]. A systematic review showed that one-third of symptomatic patients with nightmares were on beta-blocker treatment [49]. Unlike pindolol, which decreases REM sleep time, acebutolol rather increases this time, so it has been concluded that sympathomimetic activity is not involved in the pathogenesis of sleep disorders [45].

### 4.4. Hallucinations and Delirium

Hallucinations are a mental distortion manifested in the form of imaginary sensory experiences that seem real to the patient experiencing them [50]. Current data regarding the relationship between beta-blockers and hallucinations are limited. Several studies have shown that lipophilic beta-blockers may have a causal role, with the remission of symptoms after the cessation of treatment. Goldner et al. showed that there is a relationship between metoprolol and visual hallucinations, a side effect generally attributed to dreams or nightmares, consequently underdiagnosed. At the same time, patients are reluctant to report such symptoms. Therefore, it is difficult to estimate a real incidence of hallucinations [51]. Sirois et al. reported that hallucinations occur in patients treated with metoprolol as an isolated symptom, but in elderly patients or those with preexisting cognitive deficits may degenerate into delirium [52]. Hallucinations caused by metoprolol usually disappear a few days after stopping the treatment [51].

Delirium is a clinical syndrome characterized by pathological disturbance of consciousness and perception manifested through hallucinations, delusions, or illusions [53]. Beta-blockers can cause delirium, especially in the elderly or those with preexisting cognitive dysfunction [54]. Katznelson R et al. showed that beta-blocker treatment is associated with doubling the risk of delirium after vascular surgery [55]. Harrison et al. demonstrated that patients who receive beta-blocker treatment had a significantly higher incidence of delirium compared to patients treated with calcium channel blockers [56]. Moreover, a directly proportional relationship between the beta-blocker dose and the risk of delirium-compatible symptoms has been reported. The incriminated mechanism is related to the antagonistic action of beta-blockers on the serotonin-sensitive adenylate cyclase system [57]. Psychosis, usually in the context of delirium, has been reported rarely in patients treated with propranolol, metoprolol, and atenolol [57,58,59].

### 4.5. Parkinson’s Disease

Parkinson’s disease is the second most common neurodegenerative disease after Alzheimer’s disease, but the fastest growing in its category with a worldwide increase in the number of patients with the disease in recent decades [60]. Alpha-synucleine, a protein encoded by the SNCA gene, is the major constituent of Lewy bodies deposited in brain tissue [61]. Deposition of Lewy bodies in the brain tissue is the main pathogenic mechanism involved in the dementia of patients with Parkinson’s disease [62]. A human cell model that used neuroblastoma showed that beta-adrenergic agonists suppress SNCA gene expression, in contrast to beta-adrenergic antagonists that overexpress this gene and thus increase alpha-synucleine concentration [63]. Currently, the association between Parkinson’s disease and beta-blockers is considered a causal relationship between dose and duration of treatment [64]. The potential risk of beta-adrenergic antagonists is similar to the risk of pesticide exposure and overlaps with the most common genetic determination of Parkinson’s disease [65,66]. Patients treated with propranolol have been shown to have a reduced risk of developing Parkinson’s disease. For this reason, most clinicians do not change their therapeutic strategy [67].

### 4.6. The Risk of Falling

In the aging population, fall incidents form a growing healthcare problem [68]. Among patients over the age of 65, one out of three has at least one fall each year [69]. Falls significantly increase morbidity and mortality; they are associated with reduced quality of life and lead to increased health care costs [70]. One of the risk factors for falls is the use of certain drugs, including beta-adrenergic antagonists [71]. The mechanisms by which beta-blockers may increase the risk of falls refer to the induction of bradycardia, reducing the cardiac output, hypotension, and associated dizziness [72]. Pharmacological effects and adverse events may vary between beta-blocking agents, mainly related to the degree of lipid solubility, intrinsic sympathomimetic activity, and cardioselectivity [72,73]. A meta-analysis that included 2917 patients showed that treatment with non-selective beta-blockers significantly increased the risk of falling in elderly patients, in contrast to cardio selective beta-blockers that did not involve this risk. Ham A.C. et al. argued that the fall risk should be considered when prescribing a beta-blocker in the age group, and the pros and cons for beta-blockers classes should be taken into consideration [74].

## 5. Central Nervous System Therapeutic Effects

### 5.1. Migraine

The migraine, characterized by recurrent episodes of moderate–severe headache lasting 4–72 h, is a common disease that affects 15% of the general population [75,76]. Prophylactic therapy is aimed at patients with at least two episodes lasting more than 24 h/month and side effects despite the acute treatment of the headache [77]. Data from randomized clinical trials show that beta-blockers, in particular propranolol, metoprolol, and timolol, are effective in the prevention of chronic migraine. For hypertensive, non-smoking patients under the age of 60, beta-blockers are a reasonable option as first-line prophylactic medications [78]. A recently published meta-analysis evaluating the efficacy of various therapeutic classes in the management of vestibular migraine (antiepileptics, calcium channel blockers, beta-blockers, and serotonin reuptake inhibitors) showed that beta-blockers were associated with the best improvement regarding the symptoms [79].

### 5.2. Tremor

The tremor is defined as the involuntary, rhythmic, and oscillating movement of a body segment [80]. Essential tremor, the most common cause of acting tremor in adults, can be partially treated symptomatically. Propranolol monotherapy has been shown to be effective. Doses of 120–240 mg/day may reduce the severity of limbs tremor associated with essential tremor [81]. According to the results of randomized clinical trials, propranolol may reduce tremor amplitude by approximately 55%, measured with the accelerometer. A clinical trial showed that the association of primidone reduced the tremor amplitude by 60%, thus providing additional benefits [82]. The dual combination of propranolol and primidone has been shown to be more effective than monotherapy in the treatment of postural and kinetic tremor [83].

### 5.3. Akathisia and Alcohol or Benzodiazepine Withdrawal Syndrome

Akathisia is a clinical syndrome characterized by the feeling of inner discomfort and an imperative need for movement [84]. Acute akathisia most often occurs as a side effect of drugs, such as neuroleptics, selective serotonin reuptake inhibitors, or calcium channel blockers. Symptoms with a prolonged duration over 3 months characterize chronic forms, late akathisia, or abstinence [85]. Treatment of akathisia is difficult. Prevention and proper dose adjustment for drugs with this side effect are the most important strategies. Propranolol is the first line of treatment for acute forms of akathisia [86]. Atenolol, by reducing the symptoms of autonomous hyperactivity, is only an adjuvant measure in patients with alcohol or benzodiazepine withdrawal syndrome, with no proven benefit over delirium or seizures [87].

### 5.4. Anxiety and Posttraumatic Stress Syndrome

Propranolol was the first anxiolytic generally studied, but the interest has declined as selective serotonin reuptake inhibitors have become the first-line pharmacological treatment across the range of anxiety disorders [88]. Propranolol selectively blocks protein synthesis, thereby prohibiting the ‘reconsolidation’ of the fear memory while sparing declarative memory [89]. A meta-analysis suggests that propranolol has potential for the treatment of anxiety disorders that are rooted in the presence of disturbing memories, particularly posttraumatic stress disorder [90]. The administration of propranolol in the first 6 h after trauma significantly reduces the risk of developing posttraumatic stress disorder [91,92]. Propranolol is currently considered an amnesic agent used to reduce traumatic memory rather than a general anxiolytic [93].

Treatment with atenolol in association with scopolamine (antimuscarinic agent) has been proposed as a fast-acting anxiolytic, useful in the management of acute anxiety, especially before medical procedures (dental, minimally invasive) [94]. Recent data from a retrospective study suggest that atenolol may achieve the control of the symptoms in patients with anxiety or posttraumatic stress disorder, with reported efficacy of up to 86% at doses of 100 mg/day. The main advantage is the better tolerability to propranolol, but the disadvantage is that, so far, there are no randomized clinical trials regarding this topic [95].

### 5.5. Aggression

Aggression is a motor and visceral social behavior that consists of directing aggressive stimuli, physical or mental, of attacking an individual or for defense. Overall, the evidence for any successful treatment of aggression with any agent, or class of agents, is limited. Beta-blockers appear to be a well-tolerated class of drugs in patients with aggression related to traumatic brain injury [96]. Furthermore, beta-blockers appear to be effective in reducing aggression among patients with a variety of neuropsychiatric conditions, such as schizophrenia, dementia, or behavioral disorders [97,98,99].

### 5.6. Obsessive–Compulsive Disorder

Obsessive–compulsive disorder is characterized by intrusive and inappropriate obsessions accompanied by repetitive behavioral or mental compulsions [100]. Specific treatment with selective serotonin reuptake inhibitors or tricyclic antidepressants is effective in 80% of cases, and 20% of them have a partial response [101]. Pindolol, although it has antagonistic action on presynaptic 5-hydroxytryptamine 1A receptors, may have serotonergic effects due to its sympathomimetic activity [102]. Pindolol amplifies the effect of selective serotonin reuptake inhibitors and thus has been used adjunctively to enhance the benefits of selective serotonin reuptake inhibitors in obsessive–compulsive disorder and panic disorders refractory to standard therapy [101,102,103].

## 6. Limitations and Strengths

This paper has several limitations. First, the use of non-selective and lipophilic beta-blockers might decrease over time, also reducing the associated side effects. Second, the outcomes reflecting neuropsychiatric disorders will be underestimated because of the reduced addressability of the patients to a specialist. Third, the therapeutic effect of beta-blockers in psychiatric conditions is hard to prove through randomized clinical trials because there are already first-class recommendations of guidelines for other drugs than beta-blockers in different pathologies. However, data from observational studies offer hope regarding this topic. The development of the current society has determined an increase in the number of patients with psychiatric disorders, and through this article, we draw attention to the use of cardiovascular medication in this special category of patients. The main strength of this narrative review is to highlight the pros and cons of using beta-blockers, where lipophilia plays a leading role. Thus, beta-blockers are recognized as having an integrative role both in neuropsychiatry and cardiovascular specialties. As far as we know, this is one of the few articles that includes such extensive research in multiple databases regarding the lipophilicity of beta-blockers and their neuropsychiatric side effects.

## 7. Future Perspectives and Conclusions

Beta-blockers are a class of drugs with important benefits in cardiovascular pathology, mainly by reducing mortality. The number of patients with psychiatric disorders is increasing, and the elderly are one of the most affected categories. Thus, for the recommendation of beta-blockers in various pathologies (migraine, tremor, akathisia, anxiety, aggression, or obsessive–compulsive disorder), there are both pros and cons. Starting from these neuropsychiatric consequences (fatigue, depression, sleep disorders and nightmares, hallucinations, delirium, Parkinson’s disease, or the risk of falling), we suggest to the clinicians to apply special attention to the pharmacological characteristics of the different agents in the class. Hydrophilic beta-blockers may be an option in selecting the appropriate beta-blocker for the elderly patient, but further studies are needed to accurately identify patients at high risk of side effects on the central nervous system. However, the therapeutic effect of some lipophilic beta-blockers in the management of neuropsychiatric disorders is not negligible. At the same time, it is important to use beta-blockers in well-established and correct indications, in appropriate doses for each patient, and to monitor side effects throughout the treatment. Further studies regarding the benefits of lipophilic beta-blockers are needed. Thus, competent European and worldwide committees could take them into account and maybe include all the above in a guideline as a higher-class recommendation. Moreover, this paper is a place to start for researchers in the field, highlighting the fact that not only the type of beta-blocker matters but also the dosage, all being individualized for each patient profile.

## Figures and Tables

**Figure 1 medicina-57-00155-f001:**
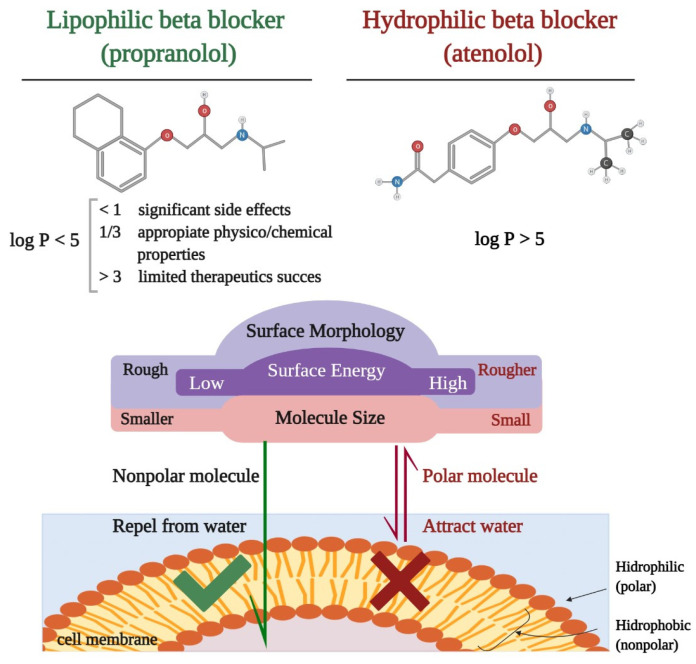
Schematic representation of the difference between lipophilic and hydrophilic beta-blockers: chemical formula, partition coefficient (log P), physical properties, and permeability through a biological membrane.

**Figure 2 medicina-57-00155-f002:**
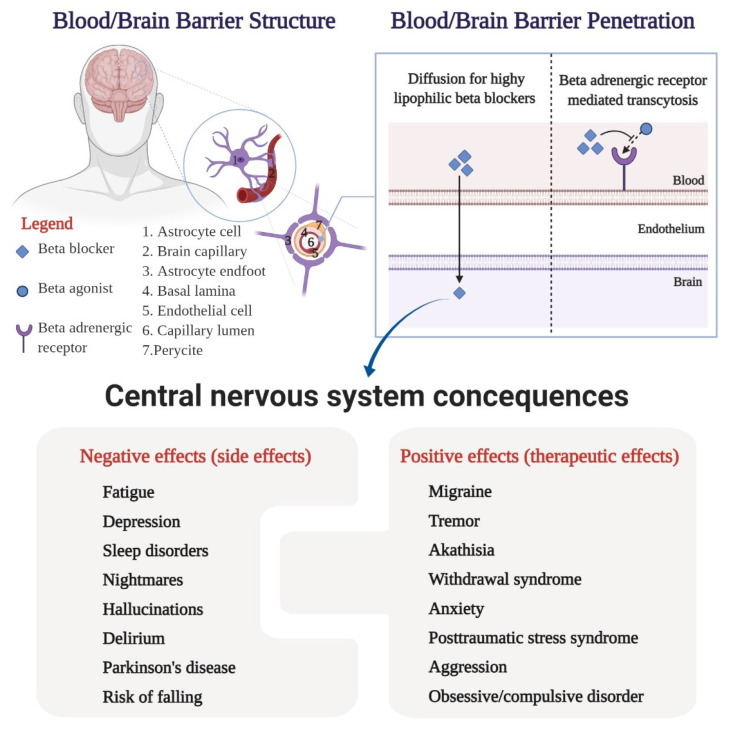
Central nervous system consequences: angioarchitecture of the blood/brain barrier, delivery of the drugs to the brain, and interconnected negative and positive effects of the beta-blockers.

**Table 1 medicina-57-00155-t001:** Beta-blockers classification regarding neuropsychiatric effects.

		High Lipophilicity	Moderate Lipophylicity	Low Lipophilicity or Hydrophile
**With ISA**	Beta-1 selective	Pindolol Penbutolol	Acebutolol Betaxolol	Carteolol
	Non-selective			Labetalol
**Without ISA**	Beta-1 selective		Metoprolol Bisoprolol Nebivolol	Atenolol Esmolol
	Non-selective	Propranolol Timolol	Carvedilol	Nadolol Sotalol

Legend: ISA = Intrinsic Sympathomimetic Activity.

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
