# Peer review of "Neuropsychiatric Consequences of Lipophilic Beta-Blockers"

_medicina, 2021, doi:10.3390/medicina57020155_

Round 1

Reviewer 1 Report

The topic in this paper is of great relevance to the majority of physicians, since use of beta blockers crosses the gamut of subspecialties, and also pertains as well to scientists in each field.

The title accurately describes the content, with the abstract following with a summary.

The introduction furnishes the background, goal, and states the purpose clearly.

The presentation is orderly, logical, classical, and readable. The discussion authoritatively considers a great deal of information, reflecting a great deal of work.

There are mini-discussions with the data nicely grouped and not dominated by the authors’ opinion. authors interpretation. Perspective and placement of their work among other studies is included.

The limitations and strength paragraph would be helpful, as well as a list of key words at the beginning, and limitations and delimitation paragraph.

A central schematic or graphic illustration outlining the concept of the paper would be extremely helpful. Expansion of the captions for groups in Table1 could show the clinical correlates and perhaps key words regarding mechanism in the captions above the Table 1.

There is a clear outline of purpose and focus is maintained throughout the manuscript.

Referencing is a bit brief considering the information in the manuscript However, in an overview with details in each section might make the piece too lengthy and lose the readers’ interest. Nonetheless additional references, preferably within the last five years, would be welcome.

In conclusion there is much food for thought, and I suspect that almost every general reader will learn much here as did I.  A great strength of the paper is the fluidity and outstanding style that is not imposing while covering much territory. Yet it does not talk down to the reader. It appears the thrust of the paper is to connect with practitioners, not to write a treatise. It is exceptionally successful in this respect.

Basically, one of the interesting episodes is the descriptions that elaborate that effects of beta-blockers depend upon lipophilicity, individual characteristics of the beta blocker, activity of transport mechanisms, timing of treatment, pH, oxidation state, genetic variation, premorbid state, and other covariates.

The authors are to be congratulated on their work.

Author Response

Dear reviewer,

Thank you for all your comments.

We introduced a limitations and strength paragraph. We think it is very useful for the reader to know them.

We have introduced 2 schematic and graphic illustrations for a better understanding of the concept of the paper.

Also, we have added 6 new references, most of them from 2020.

Hope we have touched all the points you asked us to change.

If there are any other changes you consider we should make, please let us know.

Yours sincerely,

All the authors

Reviewer 2 Report

Comments

  1. Abstract can be more concise. It should be brief but comprehensive.
  2. p2, l68, the word particularities should be replaced by another suitable word.
  3. l110, what is (24)?? This is a reference citation or something else?
  4. ref37, ref39 web links can be provided if doi is not available.
  5. ref81, what is x. ??? It should be removed.
  6. There should be one or two figures which can represent neuropsychiatric consequences of lipophilic betablockers.
  7. A lot of data available in literature on the subject chosen. A more vigorous comparison involving figures, tables and graphs will make the article more attractive for readers.

Author Response

Dear reviewer,

Thank you for all your comments.

  1. We have reduced the number of words of the abstract and tried to make it more concise and comprehensive.
  2. We have changed the word “particularities” from p2 with “characteristics”. Hope it is a suitable one.
  3. (24) was a reference citation, we have changed the brackets.
  4. For ref37 and ref39 we have added the pubmed link from where the article can be accessed.
  5. We have removed the x from ref81.

6 and 7.  We have introduced 2 schematic and graphic illustrations for a better understanding of the concept of the paper.

Hope we have touched all the points you asked us to change.

If there are any other changes you consider we should make, please let us know.

Yours sincerely,

All the authors

Round 2

Reviewer 2 Report

Manuscript is revised as suggested.

Author Response

Dear reviewer, 

Thank you for your remarks, we consider them very helpful for our future work.

Yours sincerely,

All the authors